# Deep Learning: When Conventional Wisdom Fails to be Wise

## Abstract

A major tenet of conventional wisdom dictates that models should not be over-parameterized: the number of free parameters should not exceed the number of training data points. This tenet originates from centuries of shallow learning, primarily in the form of linear or logistic regression. It is routinely applied to all kinds of data analyses and modeling and even to infer properties of the brain. However, through a variety of precise mathematical examples, we show that this conventional wisdom is completely wrong as soon as one moves from shallow to deep learning. In particular, we construct sequences of both linear and non-linear deep learning models whose number of parameters can grow to arbitrarily large values, and which remain well defined and trainable using a fixed, finite size, training set. In deep models, the parameter space is partitioned into large equivalence classes. Learning can be viewed as a communication process where information is communicated from the data to the synaptic weights. The information in the training data only can, and needs to, specify an equivalence class of the parameters. It cannot, and does not need to, specify individual parameter values. As such, the number of training examples can be smaller than the number of free parameters.

## 1 Introduction

A long held form of conventional wisdom is that in order to train a model with $n$ parameters one should have at least $n$ training examples, and preferably more. The origin of this statistical "dogma" stems from linear regression and other forms of shallow learning[1]. The soundness of this dogma appears to be obvious from our experiences with linear regression: in general $n$ examples are necessary and sufficient in order to solve a system of $n$ linear equations in $n$ unknown variables. As a result, the dogma is routinely repeated and used in myriads applications of statistics to modeling data across all areas of human inquiry, often well beyond shallow learning, and to inspire a fear, if not a disgust, for the so-called over-parameterized models. The dogma is also routinely used in a variety of "back-of-the-enveloppe"calculations, for instance to infer properties or processing strategies for the human brain. Here we show, through a variety of examples, that this central dogma is valid only for shallow learning and that it is completely wrong when it comes to deep learning. Hence, in deep learning it may not be unwise to get rid of the conventional wisdom entirely.

### 1.1 The Origin of the Dogma

For the past three centuries, since the discovery of least square linear regression by Gauss and Legendre in the late 1700s (e.g. [9]), one of the most central dogma of statistics has been that a model

---

[1]The mathematically correct distinction between shallow and deep learning is whether there are hidden units/layers or not

Submitted to 36th Conference on Neural Information Processing Systems (NeurIPS 2022). Do not distribute.

should not have more parameters than data points. There is little doubt that the origin of this dogma lies in linear regression, or equivalently in linear systems of equations where in general if there are $n$ unknown variables one needs $n$ linear equations (or training examples) to uniquely solve the system. However, this is not a characteristic of linear systems alone. The same holds true immediately for logistic regression. Since the logistic function is monotone increasing, it has a unique inverse and by inverting the targets one can reduce logistic regression to a linear system. While this is true for single linear or logistic neurons, the same result holds for a shallow layer of linear or logistic neurons, since in this case each neuron operates and learns independently of all the other neurons. Similar observations can be made for single-variable polynomial regression. Thus, in short, the origin of the conventional wisdom can easily be traced back to shallow learning and basic results in linear algebra.

Not only the soundness of the dogma seems obvious from basic linear algebra considerations, but its violation in shallow learning leads to two kinds of problems: (1) an over-parameterized shallow model is not well defined, in the sense that its parameters are not uniquely determined by the data; and, as a result, (2) such a model can overfit the data by achieving low error on the training data, while performing poorly on held out data. Finally, the widespread aversion for over-parameterized models stems also from our sense of elegance and simpliciy, as embodied in the principle of Occam's razor.

### 1.2 Applications of the Dogma

While the dogma makes sense for shallow learning situations, it is often applied to deep learning situations. For instance, many articles have been published in the literature recommending that deep learning models ought to have training sets that are 10 times [1] or 50 times [2] bigger than the number of free parameters. Obviously these arbitrary, constant, and widely discording prescriptive multiplicative factors should be viewed with a grain of suspicion.

Another standard application of the dogma is to infer properties of complex, non-shallow systems, like the brain. For instance, Geoff Hinton and others like to point out that the human brain has on the order of say $10^{15}$ synapses, while human lives last on the order of $3 \times 10^9$ seconds. Assuming one training example per second, or even 1000 training examples per second, the brain does not have enough training examples to train its army of synapses. From this false premise, one may draw all kinds of conclusion from "the brain must be doing something special" to "the majority of synapses must be hardwired". However, as we shall see, all these conclusions are worthless: they may be false or true, since they are derived from a false premise. The false premise is obtained by applying a statistical principle, correctly observed in shallow learning situations, to deep learning situations.

## 2 Preliminary Evidence against the Dogma

Preliminary evidence that something may be wrong with the dogma comes from at least three directions: Bayesian statistical theory, statistical ensembles, and deep learning practice.

From a purely Bayesian perspective, selecting the complexity of a model based on the amount of training data makes no sense at all, as there is in general no relationship between the two. Using a prior that favors simple models may be convenient, or satisfy tradition, however there is no intrinsic epistemological reason for selecting such a prior. If anything, a situation with few data points may be the sign that data are hard or expensive to acquire. In turn, this is possibly the sign of an underlying complex phenomena, which may call for a complex model rather than a simple one. Using a prior that favors models with few parameters is analogous to the paradigm of searching for one's car keys at night under the only lamp present in a dark parking lot: there is no epistemoligical reason for the keys to be under the lamp. But what about Occaam's razor? As noted in [10, 11], such a prior is not needed to implement Occam's razor which naturally emerges from the Bayesian framework. To see this in a simple way, imagine having an overall class of models comprising two sub-classes of models: simple models (S) and complex models (C). Imagine that a priori one has no preference between the two classes $S$ and $C$, and likewise that within each class one has no preference among the models in that class. Let $s$ and $c$ denote the value of the constant prior probability shared by all the models in class $S$ and in class $C$ respectively. Thus the overall prior distribution must satisfy

$$s|S| + c|C| = 1$$

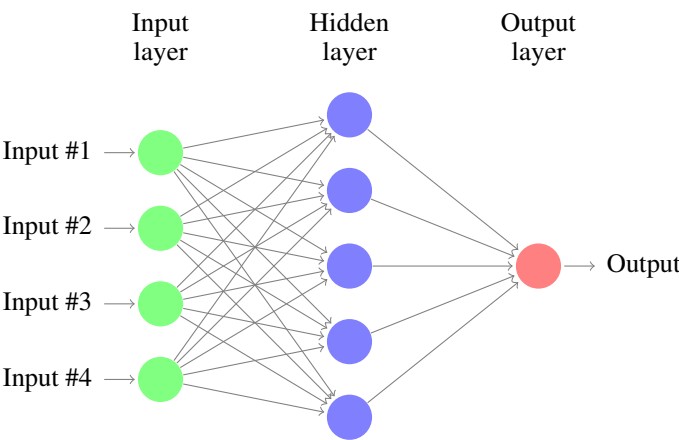

Figure 1: A $A(4, 5, 1)$ architecture.

where $|S|$ and $|C|$ represent the volumes of the corresponding classes. Because the complex models have more parameters, in general $|S| << |C|$. As a result, we must have: $s >> c$. In short, simple models will automatically have a much higher prior probability, and this effect will tend to be reflected also in the posterior probabilities.

A second line of evidence against the soundness of the dogma comes from the widespread use, and recognized effectiveness, of statistical ensembles, where many different models are combined together, for instance through a simple weighted averaging operation. This combination alone generally results in a deep overall model, even if the individual models are shallow. And even if the number of parameters of each individual model satisfies the dogma, obviously as the number of models in the ensembles is increased, there is a point where the overall model starts to violate the dogma. Perhaps surprisingly, the over-parameterization aspect of ensembles does not seem to have systematically worried statisticians.

Finally, and perhaps most importantly, it has been observed several times that in deep learning practice that over-parameterized models can work well, with no significant sign of overfitting. However, this phenomena has been used either to criticize deep learning, or is regarded as some kind of oddity or a mystery (e.g. [12, 13, 8]), possibly requiring novel strategies for combating the over-fitting curse.

Here we set out to prove why the conventional wisdom is simply wrong when it comes to deep learning. In particular we give several examples of large networks with many parameters that can be trained with far fewer examples in both the linear and non-linear cases. We consider primarily the supervised learning framework, but through the use of autoencoder architectures we show that the same basic ideas can be applied to the unsupervised, or semi-supervised, learning frameworks. At the linear end of the spectrum of models, we look at deep, fully-connected ,linear networks. At the other extreme non-linear end of the spectrum, we look at deep, fully-connected, unrestricted Boolean networks. And in the middle of the spectrum, we look at deep fully-connected networks of linear threshold gates.

**Notation:** We use the notation $A(n_0, n_1, \ldots, n_L)$ to denote a deep feedforward architectures with $n_i$ units in layer $i$, where the input layer is layer $0$ and the output layer is layer $L$ (Figure 1).

## 3 The Linear Regime

Deep feed-forward linear networks have been studied for quite some time (e.g. [4, 7, 5, 6]) in the context of least square linear regression. One of the main theoretical results is that, in the fully-connected case, the error functions of these networks does not have any spurious local minima. All the critical points where the gradient of the error function is zero are either global minima or saddle points. As a result, properly applied stochastic gradient descent will tend to converge to a global minimum. The structure of the global minima and the saddle points can be understood in terms of Principal Component Analysis (CS). Clearly, as the depth of these models is increased the number

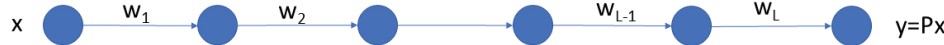

Figure 2: An $A(1, \ldots, 1)$ architecture with $L$ single-layer neurons. There is a single synaptic weight $w_i$ connecting neuron $i-1$ to $neuron i$. In the linear case, with no biases, the input-output function is given by $y = Px$ where $P$ is the product of all the synaptic weights. While the number of parameters $L$ can be arbitrarily large, a single training example is sufficient to constrain the value of the multiplier. Gradient descent rapidly converges onto an optimal solution where the product of the synaptic weight has the optimal value: $P = \alpha/\beta$ where $\alpha = E(xt)$ and $\beta = E(x^2)$ (see text).

of parameters can grow to infinity. But what are the requirements on the size of the corresponding training sets?

## 3.1 The Simplest Deep Linear Model

To begin with, we consider an architecture $A(1, 1, \ldots, 1)$, with a single linear neuron in each layer (Figure 2). For simplicity we assume that there are no biases, but the same analysis can easily be extended to the case with biases. The weights are $w_1, \ldots, w_L$ and the neural network behaves as a multiplier, in the sense that given an input $x$ the output is simply:

$$y = Px \quad \text{with} \quad P = \prod_i w_i$$

This is a deep linear regression architecture with $L$ parameters. The supervised training data consists of input-target pairs of the form $(x, t)$ that provide information about what the overall multiplier $P$ should be. Taking expectations over the training data, let $E(tx) = \alpha$ and $E(x^2) = \beta$. The error $\mathcal{E}$ is the standard least square error. It is easy to check that the error is convex in $P$ and that at the optimum one must have $\alpha - \beta P = 0$ or $P = \alpha/\beta$. It can be shown (see [3]) that, except for trivial cases, given any initial starting point, gradient descent, or even random backpropagation (feedback alignment), will converge to a global minimum satisfying $P = \alpha/\beta$.

While the architecture has an arbitrary large number of parameters $L$, in principle a single training example is sufficient to determine the value of the correct multiplier. The value of the overall product $P$ partitions the space of synaptic weights into equivalence classes: all the architectures which produce the same value $P$ are equivalent. The training data need only to provide enough information for selecting one equivalence class, but not the value of the individual weights within the equivalence class. Thus there is a manifold of equivalent solutions satisfying the optimal relationship $P = \alpha/\beta$ and the volume of this manifold grows with the number $L$ of parameters. However the training set can remain as small as a single training example, a clear violation of the dogma.

Of course, here and everywhere else in the following examples, one may wonder what could be the purpose of having $L$ layers, when a single layer could be sufficient to implement the same overall input-output function. There could be multiple purposes. The most obvious one is that the volume of the solutions grows with the depth of the architectures and this may facilitate learning. But in addition, one must also think about the possible constraints that may be associated with physical neural systems, as opposed to the virtualized simulations of neural systems we routinely carry on our digital computers using the likes of Keras, PyTorch, and TensorFlow. For example, even in the simplest linear case described above, imagine that the overall desired multiplier is $P = 2^{10} = 1024$

but that the individual synaptic weights connecting one neuron to the next are bounded in the $[-2, +2]$ range. Then no architecture with less than 10 layers is capable of implementing the optimal input-output function. Deeper architectures are needed to implement the overall optimal function and to robustly distribute the load across multiple synapses.

## 3.2 Deep Linear Models with No Bottlenecks

At first sight, one may tempted to think that the example above is due to the fact that there is a single neuron per layer. However, this is not the case and exactly the same phenomena is observed for a linear regression architectures of the form $A(n, n, \ldots, n)$ where all the layers have size $n$ and the weights are given by matrices $W_1, \ldots, W_L$. Again, in vector-matrix form, the input output relationship is given by:

$$y = Px \quad \text{with} \quad P = W_L W_{L-1} \ldots W_1$$

Again it is easy to see that this architecture has $Ln^2$ parameters. The overall input-output function corresponds to a single $n \times n$ matrix $P$. But in order to specify such a linear map, we only need to specify the images of the canonical basis of $\mathbb{R}^n$, in other words, $n$ training examples in general position are sufficient, again violating the dogma.

Note that this property remains true if the architectures also contains expansive hidden layers of size greater than $n$, or if the input and output layers have different sizes and all the hidden layers have size greater than the input and output layer (i.e. the hidden layers do not affect the rank of the optimal overall input-output function).

## 3.3 Deep Linear Models with Bottlenecks

In the previous two examples, all the layers have the same size, or are expansive. However it is easy to relax this assumption and consider compressive architectures. To begin with, consider a purely linear compressive autoencoder architecture of the form $A(n, m, n)$, with $m < n$ (Figure 3). In this case, the bottleneck layer imposes a rank restriction on the overall transformation. It is well known [4] that not only the quadratic error function of such an autoencoder has no spurious local minima, but all its critical points correspond, up to changes of coordinates in the hidden layer, to projections onto subspaces spanned by eigenvectors of the data covariance matrix. The global minima is associated with Principal Component Analysis using projections onto a subspace of dimension $m$. Obviously one can include additional linear layers of size greater or equal to $m$ between the input layer and the bottleneck layer, or between the bottleneck layer and the output layer, arbitrarily increasing the total number of parameters, but without affecting the essence of the optimal solution. The minimal training set to specify the optimal solution consists of $m$ vectors of size $n$ to specify the project hyperplane, providing another egregious violation of the dogma. Again there are large equivalence classes of parameters associated with the same overall performance (e.g. in the linear case with a single bottleneck, we have $P = AB = ACC^{-1}B$; thus the overall map $P$ is defined up to invertible transformations applied to the hidden layer). The results in [4, 7] show that the same observations can be made for arbitrary fully connected deep linear architectures (i.e. beyond autoencoders) and not only in the real-valued case, but also in the complex-valued case [6].

All the previous examples correspond to linear networks. Thus one may be mislead to think that the analyses apply only to linear networks. Next we show that exactly the same phenomena can be observed in non-linear deep architectures. Among the non-linear model to be discussed, we will examine first the most non-linear model of all which is the unrestricted Boolean model, where each neuron implements a Boolean function, with no restrictions on the kinds of Boolean functions. An unrestricted Boolean neuron with $n$ inputs implements a function $f$ with $2^n$ parameters, since one must specify one binary value for each of the $2^n$ possible entries of the truth table of $f$. Then we will consider also the case of Boolean neurons implemented by linear threshold functions, or perceptrons.

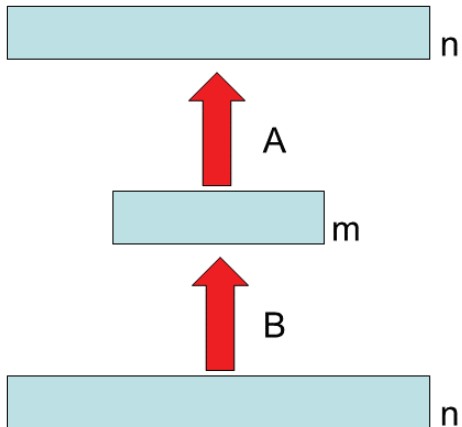

Figure 3: An $A(n, m, n)$ compressive ($m < n$) autoencoder architecture. In the linear case, the transformations $A$ and $B$ correspond to matrices and the overall linear transformation $P$ is given by: $y = Px = ABx$.

## 4  The Non-Linear Regime: Unrestricted Boolean Model

### 4.1  The Simplest Deep Non-Linear Model

We can use the same architecture $A(1, \ldots, 1)$ as in the first example above. In the Boolean unrestricted model, each Boolean function from one neuron to the next is either the identity or the negation (Boolean NOT function). So there is one binary degree of freedom associated with each layer and again the number of degrees of freedom grows linearly with the depth. The overall input-output function is either the identity, or the negation, and a single training example is sufficient to establish whether the overall function ought to be the identity or the negation of the identity. If the architecture contains an even number of negations the overall input-output function is the identity, and if the architecture contains and odd number of negations, the overall input-output function is the negation. Thus again the dogma is violated.

To get a slightly more interesting non-linear example, we can use the same architecture $A(1, \ldots, 1)$ as in the first example above, with $L$ weights $w_1, \ldots, w_L$. The difference is that all the neurons have a non-linear activation function $g(x) = x^2$ (more generally we could use for instance $g(x) = x^k$). Thus the overall input-output function is given by:

$$y = (w_L.....(w_2 w_1 x)^2))^2......)^2 = w_L^2 w_{L-1}^4 \ldots w_1^{2L} x^{2^L}$$

or

$$y = Px^{2^L} \quad \text{with} \quad P = \prod w_i^{2L-2i+2}$$

Thus in this case the multiplier $P$ realized by the architecture is positive. Again the number of parameters is $L$ and it can be arbitrarily large. As in the linear case, a single training example of the form $(x, t)$ is sufficient to determine the multiplier $P$, with a manifold of equivalent solutions corresponding to parameters satisfying $P = \prod w_i^{2L-2i+2} = \alpha/\beta$, with this time $\alpha = E(tx^2)$ and $\beta = E(x^4)$, when $\alpha > 0$. If $\alpha < 0$, the optimum is obtained for $P = 0$ which can be achieved by having at least one of the weights of the architectures equal to zero. In short, in both examples treated in this subsection, the dogma is again violated.

## 4.2 Deep Non-Linear Models with No-Bottlenecks (Unrestricted Boolean)

Consider an architecture $A(n_0, \ldots, n_L)$ where each neuron can implements any Boolean function of the neurons in the previous layer. The error function is the Hamming distance between target and output vectors. For simplicity, let us first assume that all the layers have the same size $n$. The overall input-output function is a Boolean map from $\mathbb{H}^n$ to $\mathbb{H}^n$, where $\mathbb{H}^n$ denotes the $n$-dimensional hypercube. This architecture has $Ln2^n$ parameters, since each unrestricted Boolean neuron with $n$ inputs has $2^n$ free parameters. The overall input-output map can be specified using only $n2^n$ examples. It can easily be implemented with 0 error through a large class of equivalent networks. As the number of layers $L$ goes to infinity the number of parameters goes to infinity, while the number of required training examples remains fixed and is determined entirely by the size of the input and output layers. This can easily be generalized to a Boolean unrestricted architecture of the form $A(n_0, \ldots, n_L)$, as long as there are no bottleneck layers. In such an architecture, the total number of parameters is given by: $\sum_{i=1}^{L} n_i 2^{n_{i-1}}$. The number of necessary and sufficient training examples needed to specify the overall input-output function is given by: $n_L 2^{n_0}$, and thus again the dogma is violated. The case with bottle-neck layers is treated below.

## 4.3 Deep Non-Linear Models with Bottlenecks (Unrestricted Boolean)

For simplicity, consider first an unrestricted Boolean compressive autoencoder with architecture $A(n, m, n)$ and $m < n$. The error function is the Hamming distance between the input vector and the output vector. The hidden layer can have $2^m$ states. Thus if the number of training examples is at most $2^m$, it can be realized by the architecture with 0 Hamming distortion, since every input can be mapped to a unique hidden representation and the corresponding representation can be mapped back to the same input using unrestricted Boolean gates. Obviously if additional layers of size at least $m$ are added between the input layer and the hidden layer, or between the hidden layer and the output layer, the number of parameters can be arbitrarily increased, while maintaining the same fixed training set and the ability to implement it exactly with no Hamming distortion. Thus in this regime the dogma is again violated.

In the more interesting regime where the number of training examples exceeds $2^m$, then there must be clusters of training examples that are mapped to the same hidden representation. It is easy to see that for optimality purposes the corresponding representation must be mapped to the binary vector closest to the center of gravity of the cluster, essentially the majority vector, in order to minimize the Hamming distortion. Thus, in short, in this regime the optimal solution corresponds to a form of optimal clustering with respect to the Hamming distance with, in general, $2^m$ clusters. As a back of the enveloppe calculation, assuming the clusters are spherical, these can be described by providing two points corresponding to a diameter. Thus in principle a training set of size $2 \times 2^m = 2^{m+1}$ could suffice. The number of parameters of the architecture is given by: $m2^n + n2^m$ which far exceeds the number of training examples. And even without the assumption of spherical clusters, it is clear that the number of parameters far exceeds the number of training examples, and that the gap can be made as large as possible, just by adding additional layers of size at least $m$ between the input and the hidden layer, or the hidden layer and the output layer. Thus again the dogma is grossly violated.

Finally, we turn to deep non-linear architecture where the neurons are linear or polynomial threshold gates. Linear threshold neurons, or perceptrons, are very similar to sigmoidal (e.g. logistic) neurons.

## 5 The Non-Linear Regime: Linear or Polynomial Threshold Gates

Here each neuron in the architecture is a linear or polynomial threshold function of degree $d$. In the linear threshold case ($d = 1$), any neuron with $n$ inputs $x = (x_1, \ldots, x_n)$ produces an output equal to $\text{sign}(\sum_i w_i x_i)$ in the -/+ case; or $H((\sum_i w_i x_i))$ in the 0/1 case, where $H$ denotes the Heaviside function. Such a neuron has $n$ synaptic parameters. In the polynomial case of degree $d$, the output of a neuron has the form $\text{sign}(p(x))$ in the -/+ case; or $H(p(x))$ in the 0/1 case, where $p(x) = p(x_1, \ldots, x_n)$ is a polynomial of degree $d$. The number of parameters of a polynomial threshold neuron increases accordingly. As usual a bias can also be added or, equivalently, one of the input variables is considered to be constant and equal to 1.

### 5.1 The Simplest Deep Non-Linear Model with Linear or Polynomial Threshold Gates

We can use the same architecture $A(1, \ldots, 1)$ as in the first example above. Linear or polynomial threshold neurons can realize the identity and the negation, depending on whether the corresponding incoming weight is positive or negative. So the result here is similar to the Boolean unrestricted case. For instance for linear threshold gates, without the bias, the number of parameters is equal to $L$. The number of negative weights determines how many negations are present in the chain. A single input-ouput example determines whether the overall chain should be the identity or the negation. Thus again the dogma is violated.

### 5.2 Deep Non-Linear Models with Bottlenecks (Linear or Polynomial Threshold Gates)

We can again start with a compressive autoencoder architecture with shape $A(n, m, n)$ and $m < n$ and linear threshold neurons with the Hamming error function. In the most interesting case where the number of examples exceeds $2^m$, then the optimal solution corresponds to the optimal approximation to the optimal Hamming clustering that can be achieved using linear threshold gates. The number of parameters of this architecture is $2nm$ which is not necessarily less than the number $2^{m+1}$ of required training examples, under the spherical cluster assumption. However, as in the similar previous examples, the number of parameters can be increased arbitrarily by adding additional layers of size at least $m$ between the input and the hidden layer, or between the hidden layer and the output layer. Thus once again there are large equivalence classes in parameter space (e.g. applying permutations to the neurons in a given layer) and the dogma is grossly violated.

## 6 Discussion

The conventional dogma that models ought to have less parameters than the number of training examples is a mere product of shallow learning. It arises, and should be applied, only in shallow learning situations. As soon as one moves to deep learning situations, the dogma becomes non-sense and all the expectations it creates are simply wrong, *even in the linear case*. It is simply time to think about deep models in a different way, without the expectation that over-parameterization must necessarily lead to over-fitting. This is not to say, of course, that over-parameterized deep learning models cannot overfit, but expecting them to do so just because they are over-parmaterized is unwise and unnecessary.

Over-parameterized models tend to partition the parameter space into large equivalence classes. All the parameter settings within one class are equivalent in terms of overall performance. Neural learning can be viewed as a communication process where information is communicated from the training data to the synaptic weights. The training data needs to contain enough information to select one of the equivalence classes, but not any particular setting of the weights within that class. Thus the information needed to specify one equivalence class is much less than the information required to specify a particular setting of the weights. And this explains why the number of data points can be much less than the number of parameters. Furthermore, the structure of the deep models and the partitioning into equivalence classes is such that it is not even possible for the training data to be able to specify each individual weight of the architecture. This is because the system cannot distinguish between two different settings of the parameters within the same equivalence class. For instance, once the optimal class is achieved with a particular setting of the weights, the gradient of the error is zero and there is no way of exploring or distinguishing other optimal architectures in the same equivalence class.

Shallow learning, in particular linear regression, already contains many of the central themes of machine learning: from the use of a parameterized family of models, to model fitting by error minimization, to prediction and so forth. However, when transitioning to deep learning, linear regression is misleading in three major aspects. First, it has an analytic closed-form solution. Second, it is interpretable (or visualizable, at least in low dimensions). Third, it requires that the number of training examples be equal or even exceed the number of parameters in order to completely determine the solution. The first two points are now well established and accepted. We use stochastic gradient descent for deep learning model fitting and almost no one cares about not having a closed-form analytic solution. Likewise, no one expects to be able to easily visualize complex non-linear surfaces in high-dimensional spaces, although many are still working on various other issues related to

interpretability. However, we are still struggling with the third point. It is time to move on this front too.

It should be clear from the examples presented that one of the emergent characteristics of over-parameterized regimes is the existence of large equivalence classes in parameter space, all associated with roughly the same level of overall performance. The training data needs only to provide enough information to select one of the equivalence classes (at the relevant quantization level), and not to specify the value of each one of the parameters. Reflecting back on the human brain, most mature human brains can pass the Turing test and achieve some form of general intelligence using architectures that are similar, at least at the macroscopic level, and at the level of the basic hardware components (e.g. pyramidal cells), but presumably with significant differences at the level of individual synapses.

Finally, there is the question of when deep architectures overfit the data. The results presented here provide a clear answer. Consider an architecture with $w$ parameters. At the proper level of quantization of the weights and the error function, the architecture may partition the space of weights into $e$ equivalence classes. Thus $\log_2 e$ bits are needed to specify one of the equivalence classes. If the training data provides less than $\log_2 e$ bits of information, then it does not contain enough information to select a relevant equivalence class and overfitting may occur. If the training data provides $\log_2 e$ bits of information to select an equivalence class, then there is no overfitting and providing more data is not necessary. In the case of a classification architecture with independent binary inputs of length $n$, $k$ training examples contain on the order of $kn$ bits of information. Thus the important question is not whether $k \approx w$ (conventional wisdom) but whether $kn \approx \log_2 e$.

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
