# OpenReview forum: "Deep Learning: When Conventional Wisdom Fails to be Wise"
_NeurIPS.cc/2022/Conference — NeurIPS 2022 Submitted_

### Official Review · Reviewer_6YoB · 2022-07-07

**Rating:** 2
**Confidence:** 5
**Soundness:** 4 excellent
**Presentation:** 1 poor
**Contribution:** 1 poor

**Summary:**

In this work, the authors explore the question of why overparameterized models learn well, despite the fact that classical/traditional statistics knowledge suggests that having more model parameters than datapoints is harmful for generalization. They develop a few cases where overparameterized models provably generalize well:

* Deep linear networks
* Deep boolean networks
* Deep non-linear networks

In some cases, the optimal number of parameters can be shown to be greater than the number of training datapoints.

**Questions:**

Do any of the "dogma violating" situations in the paper have different properties from the ones already reported in the literature?

**Limitations:**

Limitations and impact adequately discussed.

**Strengths And Weaknesses:**

The strongest part of the paper is the explicit construction of some of the examples; they would be very good for lectures/problem sets in introductory ML courses due to their simplicity.

However, overall the phenomenology of overparameterized networks and generalization has been well studied, particularly in the last few years. The literature on "double descent" is one major example. See papers like:

https://www.sciencedirect.com/science/article/pii/S0893608020303117
https://proceedings.mlr.press/v119/adlam20a.html

Papers like this investigate some of the root causes of the success of overparameterization, both empirically and theoretically. This literature has not been engaged with in the paper.

Simply developing some specific examples where overaparameterized models have good generalization is not a novel or impactful contribution to the field. This phenomenology has been known since the advent of deep learning, and as discussed above the origins of this phenomenon have been very well (and quantitatively) studied in more detail than this paper.

---

### Official Review · Reviewer_38Gz · 2022-07-10

**Rating:** 2
**Confidence:** 5
**Soundness:** 1 poor
**Presentation:** 2 fair
**Contribution:** 1 poor

**Summary:**

In this paper, the authors stipulate that a transition from deep to shallow learning **necessitates** discarding the practice of having a larger training set over the number of model parameters. The authors state that this practice is derived from the statistical learning theory work based on linear models.

The authors then proceed to justify this point through a set of general inspections of simplified deep linear & non-linear models.

**Questions:**

N.A.

**Limitations:**

- As noted in the **Weaknesses** section above, the authors do NOT provide a thorough argument for their claim that a transition from deep to shallow learning necessitates the obvious choice of residing in the over-parameterized regime.

- The figures provided are all too rudimentary and add little value to the work itself

- There exists no formal theoretical results on the rate of convergence to a global optima w.r.t. training time and a subsequent analysis of whether attaining such a solution seems feasible under practical settings

- There already exist long-established works on linear models, showcasing the phenomenon of double-descent in the test error. As to how the authors formally unify that seemingly contradictory viewpoint with their claims here

[1] Statistical mechanics approach to early stopping and weight decay, *S. Bös*

[2] Generalization ability of perceptrons with continuous outputs, *S. Bös, W. Kinzel, and M. Opper*

[3] Avoiding Overfitting By Finite Temperature Learning and Cross-Validation, *Siegfried Bös*

**Strengths And Weaknesses:**

**Strengths**: It has been difficult to determine the exact strengths of this work.

**Weaknesses**:
1. While the aspect of over-parameterization in DNNs is yet to be fully understood and is a subject of heavy investigation over the past few years, the authors completely fail to acknowledge the vast literature already existent on this topic

2. **Most notably, the authors fail to clarify the novelty of their work over this existent literature**

3. The authors claim that the practice of working in the under-parameterized regime even for DNNs is drawn from the observation that to solve for the values of *p* network weights, one would need an equal no. of parameters. This is to me a highly misconstrued understanding, as the practice is derived from the observed bias-variance tradeoff, itself explained from learning theory perspectives.

Specifically, given the increasing test error past an optimal model complexity of shallow NNs, it was understood that a continuation of the increase in model complexity can only worsen the generalization performance. In that light, the phenomenon of double-descent was definitely surprising

4. Contrary to the authors' broad claim here, the work of [1] argues that it is still possible for over-parameterized models to perform poorly under certain training circumstances, hence a blanket argument for over-parameterization in DNNs has to be made with some caution, which is not provided in this work

5. The authors provide very limited well-founded empirical and/or theoretical results to back their main claim. For e.g. in their analysis using a highly simplified over-parameterized deep linear network, the authors argue why a single example is in principle *enough* to train the setup. This is an inaccurate claim for the following reasons:

- in an over-parameterized setup, there exists a multitude of network weight choices satisfying the global minima configuration each having a different generalization property. As to which one should be and is selected from this set of choices so as to achieve successful model training is left totally unanswered by the authors

- it is understood and as is noted, that the landscape of a deep linear model does not contain any local minima. Nevertheless, saddle points exist and given the non-linear learning dynamics of the network weights, the authors do not put forth a method of how to escape a saddle point with the widely used gradient-based optimization methods

- In the case of a noisy label, the exact fitting to the single training point is detrimental for generalization purposes and hence arguing for why over-parameterization *makes sense* in such a model regime is quite naïve

- The work fails to provide any new theoretical estimate on the amount of over-parameterization necessary, c.f. [2]

[1] Bad Global Minima Exist and SGD Can Reach Them, *Shengchao Liu, Dimitris Papailiopoulos, Dimitris Achlioptas*

[2] A Convergence Theory for Deep Learning via Over-Parameterization, *Zeyuan Allen-Zhu, Yuanzhi Li, Zhao Song*

---

### Official Review · Reviewer_bnfh · 2022-07-11

**Rating:** 3
**Confidence:** 4
**Soundness:** 3 good
**Presentation:** 3 good
**Contribution:** 1 poor

**Summary:**

The paper discusses the widely mentioned heuristic, that the number of parameters of a statistical model should not be larger than the number of training data points. By analyzing a sequence of simple “deep, layer-wise, feedforward” models the paper shows that there is no straightforward relationship between the number of parameters and the functional expressivity (complexity) of a model. Accordingly, deep models can easily be constructed where the number  of datapoints to uniquely determine the input/output function is arbitrarily lower than the number of model parameters. The main idea is that transformations involving many layers can often be collapsed into a transformation with a single layer that is appropriately constructed - the number of parameters of that single layer determines the required number of training data points. Adding more and more parameters via deeper layers then allows to arbitrarily inflate the model’s parameter count. The argument also goes through in essence when layers have different widths (by focusing on the most narrow layer), and in case of non-linearities between layers. The paper contains no empirical evaluation.

**Questions:**

**Questions and minor comments:**

It would be great to see the analysis in 5.2 for MSE (regression), not the Hamming error function. How would the clustering argument look like in this case?

The paper somewhat fails to explain why we tend to see statistical models with eye-watering parameter counts in practice. Following the construction in the paper, many of these parameters should be superfluous (any layers wider than the bottleneck should be more or less reducible to the bottleneck-width). Yet, many rounds of architecture optimization (via humans or automated procedures such as neural architecture search) have left us with models with superfluous and redundant parameters. Work such as the Lottery Ticket Hypothesis, and results in network compression (where often >90% of parameters can be removed after training, but not before) suggest that having more parameters than necessary to implement the desired functional input/output relation might be crucial for learning. This is mentioned in passing in the paper, but I think this question lies at the heart of one of the big riddles in deep learning: the question is not ‘why do we use more parameters than datapoints’ (which is not a very meaningful question as pointed out in the paper), but ‘why do we need way more parameters during training than to implement the desired final function’? I think it would be nice to add a bit more discussion along these lines to the paper.

In the mathematical examples given in the paper over-fitting is not possible. The examples simply show the required number of datapoints to determine the correct parameter equivalence class for a given model architecture. For over-fitting to be possible the discussion would need to be expanded to e.g. held-out regions in the data manifold (test-set generalization from a training set that is insufficient to fully determine the parameter equivalence class) or noisy training data. This would probably, I think, lead to a formalism similar to SLT.

Lines 68-86: Bayesian Occam’s Razor - the discussion in the paper is a bit imprecise (I personally really like the discussion and particularly the illustration in MacKay’s textbook Chap. 28, p. 344, perhaps it can serve as inspiration to overhaul the paragraph in the paper). The discussion could be more clear by associating model complexity with the ability to implement a large set of functions (regardless of the number of parameters, and regardless of the functional complexity of a single function under the model; the latter does not even have a well-defined meaning in the framework).
 * The epistemological reason to prefer a simple model over a complex one in light of few datapoints is then that a small number of datapoints can only be used to select one out of a small number of functions - a small number of possible functions is what corresponds to a simple model. In contrast a complex model can implement a larger set of functions, and selecting the right one necessarily requires more information / data points. This is not to say that the individual functions implementable by the simple model cannot have “complex functional form”. E.g. two models could be polynomials of degree 10, but the simple model only considers coefficients in $[-1, 1]$ whereas the complex model considers coefficients in $[-100, 100]$. The functional complexity of any particular function is the same under both models.
* The second sentence in line 83 (continuing to 84) contradicts the main message of the paper. The paper generally argues that the number of parameters is not related to the complexity of a model (the set of possible functions implementable by the model) - so the inequality in line 84 would not hold. Again, I think this seeming contradiction can easily be resolved by being more careful: the difference between a simple and a complex model is not the number of parameters, but the set of implementable functions under all possible parameter settings.


**Limitations:**

The paper does not address non-feedforward architectures such as e.g. ResNets and RNNs, but this is a minor detail.


**Strengths And Weaknesses:**

**Pro:**
 * The paper is very easy to read and follow and the main argument is built in logical, consecutive steps.
 * The technical arguments in the paper are correct.
 * The paper is self-contained, and does not require extensive knowledge of sub-fields of ML.

**Con:**
 * The non technical writing is highly opinionated, criticizing a perceived “dogma” dozens of times in the text (the word dogma appears 29 times in the paper). This is not a problem per se, but makes the manuscript more suitable for an op-ed or commentary article, which is a format not supported by NeurIPS.
 * The main argument is that functional complexity (expressiveness) of a model is independent of the number of model parameters - unfortunately this is never spelled out with clarity in the paper. This should be obvious after an undergrad course in statistical modeling and ML - however, I agree that the “dogma” has been fairly widely mentioned in the deep learning literature, and to this day the number of parameters is often taken as a proxy for model complexity in deep learning (e.g. in “scaling laws”).
 * The main argument above has been addressed formally in various frameworks, perhaps most relevant to the current discussion in statistical learning theory, where model complexity is measured via the VC dimension (instead of parameter count) and learnability bounds that relate the required number of training datapoints to bounds on test error (in the i.i.d. setting). This is not discussed at all in the paper.

**Verdict:**
There is some merit to pointing out the (rather obvious) fact that statistical models, particularly deep, layer-wise, feed-forward architectures can be fully determined from datasets with fewer datapoints than model parameters. This necessarily implies equivalence classes of solutions (which I think is also interesting to point out). However, the current style and tone of the manuscript are more suited for a short opinionated or commentary piece, or perhaps a peer-reviewed blog-post (which another conference recently introduced) - neither of these formats are supported by NeurIPS. I therefore currently argue in favor of rejection because of a lack of novelty, significance, and technical depth that is expected from a typical NeurIPS original research paper. Having said that, I did enjoy reading the paper (though criticizing “the dogma” could perhaps be toned down a bit), and would encourage the authors to find an appropriate outlet.

**Improvements:**
To strengthen the work and add technical depth here are some suggestions (though one advantage of the article is that it is very easy to read because it is technically simple; so the suggestions here are aimed at adding enough “meat” for a conference publication, which might not be the right format for this article after all):
 * A thorough discussion of the (non-)relationship between functional complexity (and how to measure that) and the number of parameters of a statistical model, as well as the implications for learnability and generalization. I personally think this is well addressed in statistical learning theory (SLT) for instance.
 * Going beyond an individual model (SLT), and considering families of models (such as in Bayesian learning) and model selection - which lines 68-94 hint at - I think it would be worth discussing the minimum description length framework, where it is also obvious that the description length of a model with probabilistic parameters depends in many cases only weakly on the number of parameters.
 * The main arguments in the paper also lend themselves to some empirical investigation. For instance, one conclusion would be that increasing the width of the bottleneck layer should lead to an increased number of required datapoints, whereas increasing depth (by adding layers) should have no such effect. Though the theoretical arguments in the paper are clear and sound, a simple empirical investigation could add to the paper.
 * Expand on the discussion started in lines 329-338. This is one of the most relevant questions in modern machine learning - given that the number of datapoints is insufficient to fully determine the desired input/output relationship (and thus fully specify the correct equivalence class of parameters), how can desired functional generalization be ensured and why would it be beneficial to not reduce the number of parameters (but regularize through other means instead)? Admittedly, this is beyond the main point of the paper - but building good intuitions for these questions could be very valuable for the community.

Since the paper is written in a scholarly style, discussing SLT (and potentially MDL) as suggested above could add depth to the paper. I personally would also prefer de-emphasizing the “attack of the dogma” a bit, but that is a matter of taste, not technical correctness.

---

### Official Review · Reviewer_EpQS · 2022-07-15

**Rating:** 2
**Confidence:** 5
**Soundness:** 1 poor
**Presentation:** 2 fair
**Contribution:** 1 poor

**Summary:**

The authors revisit the problem of overfitting to data set and construct an example where neural networks overfit to their training data without it affecting their performance. The authors hypothesise that this effect arises because the data only specifies parameters up to a certain equivalence class, rather than specifying all of the parameters exactly.


**Questions:**

See above.

**Limitations:**

See above.

**Strengths And Weaknesses:**

Understanding how neural networks can obtain good test error while interpolating their data is a key problem for deep learning theory and has seen a lot of activity recently, so the authors are studying a timely topic.

However, I'm afraid there are serious issues with the premise and the presentation of the article. The introduction reads as if over-parameterisation was still considered harmful in machine learning, when in reality the advantage of over-parametrization in large neural networks has long been recognised, for example in the original bias-variance trade-off paper by Neman et al. (Neural computation 4 (1), 1–58, 1992), and of course in the more recent literature on large neural networks (Neyshabur et al. '15, Zhang '16). To say that "the dogma is routinely repeated and used in myriads applications of statistics to modeling data" or to talk about a "widespread aversion for over-parameterized models" is simply not warranted, especially in the context of NeurIPS.

To go on and claim that "many articles have been published in the literature recommending that deep learning models ought to have training sets that are 10 times or 50 times bigger than the number of free parameters" is downright misleading, even if the authors managed to find two articles supporting these claims (once of which dates from 1995...)

Reading the article further, it becomes quickly clear that the authors are not aware of the enormous amount of recent work that analyses benign overfitting and the related double descent phenomenon in neural networks (see the two references above) and in random feature regression, where this effect is well-studied and understood. I am providing a few references with a focus on the theory below.

I would therefore invite the authors to revisit their results in light of the recent experimental and theoretical findings, and to resubmit their article afterwards.

- Trevor Hastie, Andrea Montanari, Saharon Rosset, and Ryan J Tibshirani. Surprises in high-dimensional ridgeless
least squares interpolation. arXiv preprint arXiv:1903.08560
- Song Mei and Andrea Montanari. The generalization error of random features regression: Precise asymptotics and
double descent curve. arXiv preprint arXiv:1908.05355
- Peter L Bartlett, Philip M Long, Gábor Lugosi, and Alexander Tsigler. Benign overtting in linear regression.
Proceedings of the National Academy of Sciences, 2020
- Deep learning: a statistical viewpoint. arXiv preprint arXiv:2103.09177, 2021.

---

### Public Comment · ~Celestine_Preetham_Lawrence1 · 2022-11-09
**Shallow learning in recurrent networks can have deep functionality [1]**

I enjoyed reading the paper, and just wanted to comment that shallow recurrent networks can also have a large number of equivalence classes (a picture similar to Figure 2 could be made).

[1] Deep feedforward functionality by equilibrium-point control in a shallow recurrent network. https://openreview.net/forum?id=401LFvBGIb

P.S. Would be nice if NeurIPS could allow a post-decision archival version of the paper (with the author name De-anonymised). There seems to be more wisdom in some rejected papers than run-of-the-mill accepted papers.

---

### Meta-Review · Area_Chair_deis · 2022-08-24

**Recommendation:** Reject
**Confidence:** Certain

**Metareview:**

The paper explores the question of why overparameterized networks can generalize well, and shows a number of theoretical examples where overparameterized networks attain good generalization.

The question explored in the paper is an important one and the presented examples have potential pedagogical value. However, there are serious issues with the premise and the presentation of the paper; all four reviewers discussed at length those issues and recommended rejection. Unfortunately the authors did not engage in discussion with the reviewers. Given the above, it is clear that the paper is not suitable for NeurIPS.

**Award:**

No

---

### Decision · Program_Chairs · 2022-09-14

Reject